# Repurposing the Antibacterial Agents Peptide 19-4LF and Peptide 19-2.5 for Treatment of Cutaneous Leishmaniasis

**DOI:** 10.3390/pharmaceutics14112528

**Published:** 2022-11-20

**Authors:** Rima El-Dirany, Celia Fernández-Rubio, José Peña-Guerrero, Esther Moreno, Esther Larrea, Socorro Espuelas, Fadi Abdel-Sater, Klaus Brandenburg, Guillermo Martínez-de-Tejada, Paul Nguewa

**Affiliations:** 1ISTUN Institute of Tropical Health, Department of Microbiology and Parasitology, IdiSNA (Navarra Institute for Health Research), University of Navarra, c/Irunlarrea 1, 31008 Pamplona, Navarra, Spain; 2Laboratory of Molecular Biology and Cancer Immunology, Faculty of Sciences I, Lebanese University, Hadath 1003, Lebanon; 3ISTUN Institute of Tropical Health, Department of Chemistry and Pharmaceutical Technology, IdiSNA (Navarra Institute for Health Research), University of Navarra, c/Irunlarrea 1, 31008 Pamplona, Navarra, Spain; 4ISTUN Institute of Tropical Health, IdiSNA (Navarra Institute for Health Research), University of Navarra, 31008 Pamplona, Navarra, Spain; 5Brandenburg Antiinfektiva GmbH, c/o Forschungszentrum Borstel, Leibniz Lungenzentrum, 23845 Borstel, Germany; 6Department of Microbiology and Parasitology, IdiSNA (Navarra Institute for Health Research), University of Navarra, 31008 Pamplona, Navarra, Spain

**Keywords:** leishmaniasis, drug repurposing, antimicrobial peptides (AMPs), peptide 19-2.5, peptide 19-4LF, drug resistance, proliferation, cytokines

## Abstract

The lack of safe and cost-effective treatments against leishmaniasis highlights the urgent need to develop improved leishmanicidal agents. Antimicrobial peptides (AMPs) are an emerging category of therapeutics exerting a wide range of biological activities such as anti-bacterial, anti-fungal, anti-parasitic and anti-tumoral. In the present study, the approach of repurposing AMPs as antileishmanial drugs was applied. The leishmanicidal activity of two synthetic anti-lipopolysaccharide peptides (SALPs), so-called 19-2.5 and 19-4LF was characterized in *Leishmania major*. In vitro, both peptides were highly active against intracellular *Leishmania major* in mouse macrophages without exerting toxicity in host cells. Then, q-PCR-based gene profiling, revealed that this activity was related to the downregulation of several genes involved in drug resistance (*yip1*), virulence (*gp63*) and parasite proliferation (*Cyclin 1* and *Cyclin 6*). Importantly, the treatment of BALB/c mice with any of the two AMPs caused a significant reduction in *L. major* infective burden. This effect was associated with an increase in Th1 cytokine levels (*IL-12p35*, *TNF-α*, and *iNOS*) in the skin lesion and spleen of the *L. major* infected mice while the Th2-associated genes were downregulated (*IL-4* and *IL-6*). Lastly, we investigated the effect of both peptides in the gene expression profile of the P2X7 purinergic receptor, which has been reported as a therapeutic target in several diseases. The results showed significant repression of *P2X7R* by both peptides in the skin lesion of *L. major* infected mice to an extent comparable to that of a common anti-leishmanial drug, Paromomycin. Our in vitro and in vivo studies suggest that the synthetic AMPs 19-2.5 and 19-4LF are promising candidates for leishmaniasis treatment and present P2X7R as a potential therapeutic target in cutaneous leishmaniasis (CL).

## 1. Introduction

Leishmaniasis is a parasitic disease causing more than 13,700 deaths annually [1] and is classified by the World Health Organization (WHO) as one of the 20 neglected tropical diseases (NTDs) [2,3]. It is considered a vector-borne disease since Leishmania parasites (the causative agents of the disease) are transmitted to humans by the bite of phlebotomine sand flies [4]. According to some studies, humans are facilitating the worldwide dissemination of this disease due to increasing migration flows, deforestation and climate change [5,6]. Currently, this disease is endemic in 92 countries [7], and more than one billion people live in those endemic regions and are at risk of infection [7].

Leishmania parasites possess two distinct morphologies, the freely living promastigotes in the vector stage and the intracellular amastigotes in mammalian hosts. The promastigotes have an elongated shape and display an active flagellar motility. Once phagocytosed by the immune cells of vertebrate hosts, the parasite losses its flagellum and becomes spherical [8]. In patients infected with *Leishmania* there are three main clinical presentations of the disease: cutaneous leishmaniasis (CL), mucocutaneous leishmaniasis (MCL) and visceral leishmaniasis (VL) [9]. The most common presentation, CL, is endemic in more than 70 countries around the world where it accounts for 700,000 to 1.2 million cases per year [9,10]. Although less common than CL, VL is the most severe presentation, ranking second after malaria in mortality rate among parasitic diseases [1,7].

There are few available drugs against leishmaniasis, with generic pentavalent antimonials being the first line of treatment [11]. However, these compounds are far from ideal due to their severe side effects and the increasing resistance developed by *Leishmania* [12]. Amphotericin B (Ampho B) and miltefosine are alternative treatments showing a higher efficacy against leishmaniasis, but they are a high-cost option compared to antimonials [3,13]. Finally, paromomycin (PM), a natural aminoglycoside antibiotic synthesized by *Streptomyces riomosus*, is more affordable and also has significant activity against *Leishmania*.

Importantly, treatment with all the available anti-Leishmania treatments is associated with numerous side effects, ranging from mild pain at the injection site to severe symptoms derived from hepatic and renal toxicity [3]. Due to these limitations and bearing in mind, the lack of an effective human vaccine against leishmaniasis, the search for new treatment strategies is of utmost importance.

Drug repurposing remains an insightful strategy within drug discovery and development processes performed to find new applications for medicines that are already used for other diseases. Therefore, drug repurposing has also emerged as an attractive strategy for neglected tropical disease drug discovery and development, including for antileishmanial treatment [14]. Currently, anti-microbial peptides (AMPs) are attractive molecules due to their high potential as therapeutic agents. AMPs are short molecules (<100 amino acids) produced by all types of living organisms including bacteria, fungi, plants, invertebrates, nonmammalian vertebrates, and mammals [15]. In many multicellular organisms and also in humans, AMPs are key components of the innate immune response where they exert a broad spectrum of biological activities, being not only capable of killing pathogens (fungi, parasites, bacteria and viruses) but also of modulating the host immune responses [16,17]. Furthermore, AMPs have, in general, low toxicity, and their target organisms are less likely to develop resistance against them than when treated with conventional drugs [18].

Several groups of AMPs, such as melittin, cecropin, cathelicidin, defensin, magainin, temporin, dermaseptin, eumenitin and histatin have been proven to have significant action against diverse *Leishmania* species, mainly acting through parasite membrane disruption. Besides, other modes of action were reported by some of these peptides including apoptosis, mitochondrial dysfunction, immune response modulation, and DNA damage [19]. There is active research focused on improving the pharmacological properties of AMPs by reducing their potential limitations such as toxicity and susceptibility to degradation by host proteases [17,20,21].

It is well known that AMPs can be fully synthetic or modified chemically. They can be replicas of the sequences available in nature, or improved versions thereof—shorter, hybrid, or amino acid substitutions [20]. In this context, a new class of synthetic anti-lipopolysaccharide peptides (SALPs) has been successfully designed and synthesized. SALPs derived from the lipopolysaccharide (LPS)-binding domain of the Limulus anti-LPS factor (LALF) and their primary sequence were extensively modified for optimal binding to the lipid A portion of LPS [22]. SALPs were shown to combine excellent selectivity for LPS, with high neutralizing activity in vitro. In addition, they were shown to efficiently protect against septic shock in murine and rabbit models of this pathology [22,23,24]. Interestingly, SALPs showed very low cytotoxicity under physiological conditions, thereby highlighting their therapeutic potential [22,23]. The antimicrobial peptides 19-2.5 and 19-4LF belonging to SALPs family are currently under preclinical tests. They have been essentially studied against bacterial infections showing remarkable activity against Gram-positive bacteria while they had modest activity against Gram-negative strains. In addition, these AMPs have an increased ability to neutralize toxins from both types of organisms, namely LPS and lipoproteins (LP) [25,26].

In the present study, the effects of both AMPs 19-2.5 and 19-4LF were studied for the first time against CL in vitro and in vivo. Our results supported the approach of repurposing both AMPs as antileishmanial drugs.

## 2. Materials and Methods

### 2.1. Compounds. Physicochemical Properties and Bioavailability

The peptides included in this study, 19–2.5, also termed Aspidasept^®^, (GCKKYRRFRWKFKGKFWFWG) and 19-4LF (GKKYRRFRWKFKGKLFLFG), were synthesized at the Borstel Research Institute with an amidated C terminus by the solid-phase peptide synthesis technique in an automatic peptide synthesizer (model 433A; Applied Biosystems, Waltham, MA, USA) on Fmoc-Rink amide resin, according to the 0.1-mmol FastMoc synthesis protocol of the manufacturer, including the removal of the N-terminal Fmoc group. The purity of both peptides was ≥95%, as determined by High-Performance Liquid Chromatography HPLC and mass spectrometry [22,27,28,29].

The properties for drug similarity were analyzed according to Lipinski’s rule of five (molecular mass of ≤500 Da, logP [logarithm of compound partition coefficient between n-octanol and water] of ≤5, H-bond donors [HBD] of ≤5, and H-bond acceptors [HBA] of ≤10) and using topological polar surface area (TPSA) values from the Molinspiration online property calculator tool kit, using the Molinspiration property calculation program (http://www.molinspiration.com/services/properties.html (accessed on 23 February 2021)). TPSA was used to calculate the percentage of absorption (% Abs) according to the equation % Abs = 109 − (0.345 × TPSA) [30]. PM, Nipajin^TM^ and ethylenediamine tetraacetic acid (EDTA) were obtained from Sigma-Aldrich (Rome, Italy). Stearic acid, cetyl alcohol, glycerol monoestearate, solid paraffin and white vaseline were purchased by Fagron (Barcelona, Spain). Liquid paraffin was obtained from Guinama (Valencia, Spain). All other reagents were of analytical grade.

### 2.2. Animals

Six-week-old female BALB/c mice were purchased from Harlan Interfauna Ibérica S.A. (Barcelona, Spain). Animals were randomly housed in groups and kept in controlled environmental conditions (12:12 h light/dark cycle and 22 °C). All the procedures involving animals were approved by the Animal Care Ethics Commission of the University of Navarra [approval number: E5-16(068-15E1) 25 February 2016].

### 2.3. Cells and Culture Conditions

*Leishmania major* promastigotes (Lv39c5) were grown at 26 °C in Schneider’s Drosophila medium (Gibco) supplemented with heat-inactivated fetal bovine serum (FBS), and an antibiotic cocktail (50 U/mL penicillin, 50 mg/mL streptomycin). To maintain their infectivity, *Leishmania* cells were isolated from infected BALB/c mouse spleen and parasites were maintained in culture for not more than five passages.

Murine bone marrow-derived macrophages (BMDMs) were obtained as previously described [31].

### 2.4. Cytotoxicity Assay

The 3-(4,5-dimethylthiazol-2-yl)-2,5-diphenyltetrazolium bromide (MTT) test (Sigma, St. Louis, MO, USA) was performed to determine the cytotoxicity of peptides on BMDMs. MTT solutions were prepared at 5 mg/mL in phosphate buffer solution (PBS), filtered and maintained at −20 °C until use. Briefly, macrophages were detached with a scraper and 5 × 10^4^ cells were seeded per well in 96-well plates and allowed to adhere for 24 h at 37 °C in a 5% CO_2_ humidified atmosphere. The culture medium was replaced by fresh medium with increasing concentrations of peptides (from 0 to 4 µg/mL) and after 72 h of incubation, 100 µg/well of MTT was added and the plates were incubated for 4 h under the same conditions. Then, 80 µL of dimethyl sulfoxide (DMSO) was added to each well to dissolve formazan crystals. The optical density (OD) was measured in a Multiskan EX microplate photometer plate reader at 540 nm [32,33] and the half-maximal inhibitory concentration (IC_50_) was calculated. The IC_50_ represents the concentration required for 50% growth inhibition of treated cells with respect to untreated cells (controls) and was obtained by fitting a sigmoidal E_max_ model to dose-response curves. The results were expressed as means (±standard deviation, SD) from two independent experiments.

### 2.5. Activity against Intracellular Amastigotes

Macrophages were seeded in 8-well culture chamber slides (Lab-Tek^TM^; BD Biosciences, East Rutherford, NJ, USA) at a density of 5 × 10^4^ cells per well in RPMI medium and allowed to adhere overnight at 37 °C in 5% CO_2_. In order to perform the infection assay, metacyclic *L. major* promastigotes isolated by the peanut agglutinin (PNA) method [34] were used to infect macrophages at a macrophage/parasite ratio of 1/20. The plates were incubated for 24 h under the same conditions until promastigotes were phagocytized by macrophages. The extracellular parasites were removed by washing with medium, and plates were incubated with fresh medium containing 19-2.5 and 19-4LF peptides at 1 µg/mL [35]. Similarly, to analyze the effect of peptides in combination with leishmanicidal drugs (PM or Ampho B), infected macrophages were exposed to a mix containing 1 µg/mL of each peptide and 50 µM of PM or 0.025 µM of Ampho B; 72 h later, cells were washed with PBS, fixed with ice-cold methanol for 5 min and stained with Giemsa stain. To determine the parasite burden, the number of amastigotes per 200 infected macrophages was counted under a light microscope. The mean number of amastigotes per infected macrophage was determined by dividing the total number of amastigotes counted by the number of infected macrophages. Three independent experiments were performed with duplicates.

### 2.6. Gene Expression Changes upon Treatment

#### 2.6.1. RNA Retrieval from *L. major* Infected BMDMs

RNA from *L. major* amastigotes was obtained as follows. BMDMs were plated with a density of 10^6^ cells per well in 24-well plates. After 24 h of incubation, macrophages were infected with *L. major* PNA (-) metacyclic promastigotes parasites, at a ratio of 20:1 (parasites/macrophage). At 24 h post-infection, non-phagocytosed parasites were washed off with PBS and infected BMDMs were further treated with the peptides (1 µg/mL) or left untreated as a control. After 48 h of treatment, the adherent BMDMs containing *L. major* amastigotes were stored in TRI reagent (1 mL per well) at −80 °C until RNA extraction.

#### 2.6.2. RNA Retrieval from *L. major* Infected Mice

Animals were infected by subcutaneous inoculation with 10^5^ infective metacyclic promastigotes of *L. major* in the base of the tail. Eight weeks after the infection, lesions of measurable size had developed and animals were incorporated into the assay gradually once lesions reached an average area of 12 mm^2^ [31]. Three treatments were evaluated: PM (used as positive control), 19-2.5 and 19-4LF, and seven mice were used in each group, including the negative control corresponding to untreated mice. Treatments, topically administered twice daily for a period of 30 days, consisted of 50 mg of cream containing 15% of PM (1.4 g/kg/day) or a 20 µL drop of the peptide solution containing 400 µg/mL of the peptide (19-2.5 or 19-4LF) dissolved in pyrogen-free saline (PFS) (0.8 mg/kg/day) [36]. PM cream was prepared as previously described [31]. After the 30-day treatment regimen, animals were kept untreated for 3 days and then they were sacrificed. Fragments from skin lesions as well as from the spleen of infected mice were aseptically removed, immersed in 0.5 mL of RNAlater solution (Invitrogen, Waltham, MA, USA) to preserve them from RNA degradation, and conserved at 2–4 °C.

#### 2.6.3. RNA Extraction and Gene Expression Analysis

Total RNA from *L. major* amastigotes was extracted following the TRI reagent manufacturer’s protocol (Sigma, St. Louis, MO, USA) and subsequently treated with DNase (Gibco-BRL). The RNA from mice tissue samples was also extracted following the TRI reagent manufacturer’s protocol (Sigma, St. Louis, MO, USA) and subsequently treated with DNase (Gibco-BRL). One microgram of total RNA from each sample of amastigotes and mice tissues was used for retrotranscription with M-MLV reverse transcriptase (Invitrogen), following the protocol of the manufacturer. Real-time PCR was performed with iQ SYBR Green supermix (Bio-Rad) in a CFX96 system from Bio-Rad, using specific primers for each gene. *Leishmania* and mouse primers used for qPCR are summarized in Table 1 and Table 2, respectively. *Glyceraldehyde-3-phosphate dehydrogenase (GAPDH)* (Table 1) [32] was used as a housekeeping gene to normalize *L. major* gene expression. For mouse genes, the *β-actin* reference gene was used to normalize the expression (Table 2) [37]. The amount of each transcript was expressed by the formula 2*^ct^*^(actin or GAPDH)^−^^*^ct^*^(gene)^, with *ct* being the point (PCR cycle) at which the fluorescence rises appreciably above the background fluorescence [38].

### 2.7. Parasite Burden Evaluation from Infected Mice

The quantification of *L. major* burden in different tissues from mice was performed by measuring mRNA levels of the 18S ribosomal gene from *Leishmania* spp. (*Lm18S*) by reverse transcription, followed by real-time PCR as previously described [38], using specific primers (see Table 1).

### 2.8. Statistical Analyses

Statistical analyses were performed using PRISM version 5.0 (GraphPad: San Diego, CA, USA). The data are presented as means ± the SD. Comparisons between the two groups were made using the two-tailed unpaired t-test. The statistical significance was determined (***, *p*  <  0.001; **, *p*  <  0.01; *, *p*  <  0.05).

## 3. Results

### 3.1. Both Peptides 19-2.5 and 19-4 LF Displayed Leishmanicidal Activities against L. major Intracellular Amastigotes

The cytotoxicity of 19-2.5 and 19-4 LF peptides on BMDMs was tested using MTT assay with increasing concentrations of the peptides. The corresponding macrophages viability was calculated compared to the untreated control. Both peptides displayed no toxicity (100% cell viability) at all the tested concentrations (from 0.5 to 4 µg/mL)**.** According to the obtained cell viability values, in further experiments, we decided to use a concentration of 1 µg/mL for the two peptides, namely four times lower than the maximum (non-toxic) concentration tested.

To investigate the leishmanicidal effect of 19-2.5 and 19-4 LF peptides, we studied their activity against the intracellular form of the parasite, the amastigote. For this purpose, macrophages were infected in vitro with metacyclic *L. major* promastigotes. After 24 h, infected macrophages were treated with the two peptides 19-2.5 and 19-4 LF at a final concentration of 1 µg/mL. The results showed that both peptides caused a significant reduction in the number of amastigotes per infected macrophage (*p* ≤ 0.001) compared to untreated cells (Figure 1).

### 3.2. 19-2.5 and 19-4 LF Reduced the Expression Levels of Genes Related to Proliferation and Drug Resistance of L. major Amastigotes

To identify the molecular mechanism involved in peptide activity, we studied the effect of 19-2.5 or 19-4LF on the expression levels of *L. major* genes related to drug resistance (*yip1* and *ABCC6*) [39,40], parasite virulence (*gp63*) [41] and cell cycle proliferation (*Cyclin 1* and *Cyclin 6*) [42]. To this end, the gene expression level was quantified by RT-PCR using mRNA purified from BMDM previously infected with *L. major* and treated with 19-2.5 or 19-4LF as a template. As shown in Figure 2, mRNA expression levels of all studied genes were downregulated in parasites exposed to 19-2.5 and 19-4LF peptides, compared to untreated parasites (control). Significant reductions (*p* < 0.01) were detected in (*Yip1*, *Cyclin 1* and *Cyclin 6*) gene expression levels when treated with 19-2.5, while *Cyclin 1* and *gp63* mRNA levels were significantly decreased (*p* < 0.05) by 19-4LF (Figure 2).

### 3.3. Both Peptides Significantly Reduced the Parasite Burden In Vivo When Topically Administered

Lipinski descriptors for bioavailability estimation were calculated for 19-4 LF and 19-2.5 using the freely accessible program MIPC (Molinspiration Property Calculator) (http://www.molinspiration.com/services/properties.html (accessed on 23 February 2021)). These descriptors identify molecular properties important for drug pharmacokinetics in the human body including their potential oral absorption [32]. Our results predicted poor oral absorption for both peptides since they violated most of the parameters of Lipinski’s Rule of Five (Table 3). Preliminary tests were performed by administering them topically in further in-vivo studies.

To investigate the therapeutic potential of 19-4 LF and 19-2.5 against *Leishmania* in vivo, we first infected mice subcutaneously and once the areas of the lesions were 12 mm^2^, we treated them topically with the drug candidates. Specifically, peptides were administered on the skin lesions of the animal twice a day for 30 days and parasite burden was determined by measuring the *Lm18S* mRNA expression levels in the skin lesion and the spleen. When compared to untreated samples (control), the parasite burden was lower in both, skin lesions and the spleen after the administration of the two peptides (Figure 3). Significant differences in parasite burden (**, *p* < 0.01) were detected in the skin lesion with 19-4LF treatment, compared to the control (Figure 3A).

### 3.4. Both 19-2.5 and 19-4 LF Modulated the Expression of Host Cell Genes In Vivo

It has been demonstrated that pro-inflammatory and anti-inflammatory cytokines play a crucial role in immunity against CL inducing resistance/protection against the parasite (Th1 pro-inflammatory cytokines) and pathogen proliferation (Th2 cytokines) [43]. Therefore, four selected inflammatory cytokine genes (*IL 12p35*, *TNFα,* IL-4 and IL-6) and some of their corresponding receptors (*TNFR1, Il4Rα* and *IL-6R*) were analyzed by quantitative RT-PCR in the skin lesion (Figure 4) and the spleen (Figure 5) of infected BALB/c mice, treated with 19-2.5 and 19-4 LF (1 µg/mL) or left untreated. Two more genes were also studied, the inducible nitric oxide synthase (*iNOS*) whose expression is induced by cytokines [44] and the cell cycle repressor gene (*CDKN1A*) [45]. Gene expression analyses in the skin lesion of infected BALB/c mice, showed an induction of Th1 cytokine (*IL12p35* and *TNFα*) and *iNOS* gene expression levels when treated with 19-2.5 and 19-4 LF peptides. A significant variation was detected for *TNFα* expression levels (*, *p* ≤ 0.05 compared to control) when the mice were treated with 19-4 LF (Figure 4). After analyzing the spleens of mice treated with peptides, significant inductions in *IL12p35* and *iNOS* mRNA levels were detected with both treatments, as well as in *TNFα* gene with 19-2.5 (Figure 5). *IL-4* and *IL-6* (Th2 cytokines) gene expression levels were not altered either in skin lesions or in spleens after the treatment with 19-2.5. In fact, the *IL-6* mRNA level augmented only in the spleen after treatment with 19-2.5 peptide (*, *p* < 0.05) (Figure 5). On the contrary, in skin lesions of 19-4LF treated mice, a decrease in expression levels of Th2 cytokines was detected. Such a reduction was very significant (**, *p* < 0.01) for *IL-6* (Figure 4), while no significant change was observed in the spleen of the 19-4LF-treated mice for both *IL-4* and *IL-6*. Regarding cytokine receptors, *TNFR1*, *IL4Rα* and *IL6R* expression levels tended to increase in both skin lesions and spleen tissues (Figure 4 and Figure 5). The treatment with 19-2.5 increased the levels of *IL-4* gene receptors in both, skin lesions and the spleen while a significant induction of the *TNF* receptor was detected in the skin. On the other hand, 19-4LF peptide was able to significantly increase the expression levels of *TNF* and *IL-6* receptors in both types of samples, skin lesions and spleens. Lastly, the expression analysis of the *CDKN1A* gene (a negative regulator of cell cycle progression genes) showed a significant increase (*, *p* < 0.05) in the skin lesions of mice treated with 19-2.5 and 19-4LF versus untreated mice) (Figure 4), while no significant changes were detected in the spleen tissues after applying the same treatments (Figure 5).

### 3.5. P2X7 Receptor Gene Expression Was Highly Reduced by Both Peptides in Skin Lesions of BALB/c Mice

It was reported that the P2X7 receptor, expressed in immune cells, has a key role in the regulation of the inflammatory response leading to infection control [46]. To test whether the expression of the P2X7 receptor was affected by the treatment with the two peptides, mRNA expression levels were quantified in the skin lesions and the spleen of BALB/c mice infected with *L. major* and treated with 19-2.5, 19-4 LF (1 µg/mL) and PM (50 µM) or untreated (control). The P2X7 receptor was significantly downregulated in the skin lesions of the mice treated with 19-2.5 and 19-4LF (*, *p* < 0.05) and PM (**, *p* < 0.01), while no significant variation was detected in the spleen tissues with the same treatments (Figure 6).

### 3.6. The Combination of Both Peptides with the Leishmanicidal Drugs Paromomycin and Amphotericin B Greatly Enhanced In Vitro Their Activity against L. major Amastigotes

Lastly, the effect of both peptides in combination with the currently used anti-leishmanial drugs PM and Ampho B was studied. The results showed that the combination of 19-2.5 (1 µg/mL) or 19-4 LF (1 µg/mL) with PM (50 µM) dramatically reduced the number of amastigotes per macrophage from 6 (PM) to 2 or 1 (*, *p* < 0.05) when using 19-2.5 or 19-4LF, respectively (Figure 7).

Similarly, the anti-parasitic effect of Ampho B (0.025 µM) was also significantly induced when Ampho B was used in combination with the peptides. In fact, there was a very significant decrease in the number of amastigotes/macrophages from 9 (Ampho B) to 2 after such a combination (***, *p* < 0.001) (Figure 7).

## 4. Discussion

The attractive biological activities of AMPs are prompting active research in the therapeutic application of these agents to combat many infectious diseases, including leishmaniasis [20]. Herein, we evaluated the leishmanicidal activity of two synthetic AMPs, 19-2.5 and 19-4LF, against *Leishmania major*, the causative agent of CL. In vitro experiments demonstrated that both peptides displayed potent amastigote-killing activity in infected macrophages when added at low concentrations (1 µg/mL). Interestingly, we proved that 19-2.5 and 19-4LF caused no toxicity in host macrophages at 4 µg/mL (>4 times the effective concentration on amastigotes).

We also showed that the treatment of *Leishmania*-infected macrophages with 19-2.5 and 19-4LF lowered the expression levels of several *Leishmania* genes implicated in proliferation (*Cyclin 1* and *Cyclin 6*), pathogen virulence (*gp63*) and treatment resistance (*yip1)* In line with our results, these genes were similarly downregulated by CPE2, a recently reported leishmanicidal compound [47]. Importantly, our results demonstrated that 19-2.5 and 19-4LF retained their anti-parasitic potential in vivo since they dramatically reduced the parasite burden in the skin lesion as well as in the spleen of infected mice. It is worth highlighting that both peptides exerted their therapeutic activity upon topical application on the skin lesions with no need for systemic administration. Previous studies have shown the application of both AMPs as skin drugs [29]. In fact, these peptides may become insensitive to the action of skin proteases after their adequate cream formulation [29]. This strategy will likely reduce toxicity concerns associated with the systemic route and it may facilitate their clinical use if one day these peptides reach the market.

Since the modulation of the immune response was one of the mechanisms of action reported for AMPs [19] we studied whether 19-2.5 and 19-4 LF peptides altered the cytokine expression profile in skin lesions and spleens of infected BALB/c mice. Although each peptide displayed its own pattern of cytokine modulatory activity, our results suggested that they caused an increase in Th1 cytokine mRNA levels (*IL-12p35*, *TNF-a* and *iNOS*) in both, the skin lesion and the spleen. Interestingly, the Th1 type immune response induction was reported to be of critical importance for the control of cutaneous leishmaniasis [48]. In contrast, in skin lesions from *Leishmania*-infected mice treated with the peptide 19-4LF, a decrease in *IL-4* and *IL-6* gene levels was detected, in agreement with the significant reduction in parasite burden in those samples. Such cytokines have been related to a non-protective Th2 response during CL in mice [48].

It has been demonstrated that *Leishmania* clearance was enhanced by the Th1-type response, in which *IL-12* expression played a crucial role [49]. Consistent with this, an increase in IL-12, TNF and NO production by macrophages was shown to be characteristic of an efficient anti-parasitic response [50]. In addition, immunotherapy with IL-12 was reported to mediate the control of *L. major* infection in BALB/c mice and was associated with wound healing promotion, parasite burden reduction and *IL-4* downregulation [43]. Recently, it was demonstrated that the overexpression of the protein kinase LmjF.22.0810 in *L. major* (cell line LmJ3OE) resulted in a phenotype with reduced virulence. Remarkably, this cell line was shown to repress the gene expression of Th2-associated cytokines (IL-4, IL-10 and arginase 1) leading to a Th1 immune response predominance (and an increase in IL12-p35 cytokine). This immune modulation likely explained the enhanced leishmanicidal ability that mice (infected with this cell line) exhibited compared to control (mice infected with *L. major wild type*) [38]. In accordance with previous studies, our findings suggested that the Th1 response induced by 19-4LF and 19-2.5, might be responsible for the parasite burden reduction in the skin lesion and in the spleen of *L. major* infected BALB/c mice. Previous studies performed to explore the effect of *Leishmania* infection on host macrophage gene expression showed a general suppression of the cytokine receptor genes (*IL-4Ra*, *IL-6R* and *TNFR1*) in conjunction with the parasite multiplication [51]. These data were consistent with our results showing significant upregulation of three receptors *IL-4Ra*, *IL-6R* and *TNFR1* in both the skin lesion and the spleen of *L. major* BALB/c infected mice treated with both peptides 19-2.5 and 19-4LF. This overexpression could be related to the parasite burden reduction enhanced by 19-2.5 and 19-4LF peptides.

We also investigated the effect of both peptides on the purinergic receptor *P2X7* gene expression. P2X7R had been associated with tissue damage in *Mycobacterium tuberculosis* infection [52], as well as in ulcerative colitis disease, where a reduction in tissue damage was detected in mice lacking P2X7 receptors [53]. Interestingly, in both the skin lesion and the spleen, our peptides 19-2.5 and 19-4LF were acting similarly to the PM. We observed that both AMPs downregulated *P2X7R* in the skin lesion of infected mice, while no significant change in the receptor gene expression was observed in the spleen [16,46,54]. However, P2X7R was reported to play a crucial role in *L. amazonensis* elimination by a leukotriene (LT) B4-dependent mechanism [55], and its absence was associated with a more severe lesion in mice infected with *L. amazonensis* [46]. P2X7 KO mice exhibited an increased Th1 inflammation during *L. amazonensis* infection, higher levels of INF-γ and a decrease in TGF-α [46]. Such a cytokine profile is related to *L. major* parasite clearance [56,57]. It is well known that the immune response against *L. amazonensis* is different from that against *L. major* and does not implicate the Th1/Th2 balance [58]. Our findings showed an activation of Th1-associated genes (*IL-12*, *TNFa* and *iNOS*) when the gene expression level of the *P2X7 receptor* was reduced in skin lesions of *L. major* infected mice. In addition, a significant increase in *IL-6* gene expression levels was observed in the spleen in accordance with studies revealing that the P2X7 receptor promoted the secretion of the inflammatory cytokine IL-6 [59,60]. Our findings supported the P2X7 receptor as a therapeutic target for cutaneous leishmaniasis.

As previously described, cyclins are proteins involved in the cell cycle control which active the activity of kinases by forming a complex with those proteins (cyclin-CDKs) [61]. Inhibitors of cyclin-dependent kinases (CDKs) have been reported as suitable candidates for the treatment of leishmaniasis [62,63,64,65]. Recently discovered compounds with leishmanicidal activity have demonstrated the ability to inhibit the expression of such cyclins in *Leishmania major* [47]. On the other hand, terbinafine resistance locus protein (*yip 1*) codifying gene, has been detected to be overexpressed in drug-resistant *Leishmania* strains [42,66,67] Since the peptide 19-2.5 showed activity by inhibiting *yip1* gene levels, it might be also considered as a potential candidate against resistant strains. Whereas the peptide 19-4LF might reduce the parasite burden of skin lesions by inhibiting the expression levels of the gene codifying GP63 (a metalloprotease and a virulence factor also involved in host cell signaling mechanisms) among other pathways [68,69].

Lastly, the combination of those peptides with two known drugs administered against CL showed a higher effect, increasing the leishmanicidal activity of such treatments alone. During in vitro assays, Ampho B and PM displayed higher activity against intracellular amastigotes when administered in combination with those peptides. It is known that the destruction of skin and severe tissue damage caused by CL can be produced by an exacerbated immune response [70]. Therefore, the control of both, immune response and parasite burden might be considered when looking for new therapeutic options. The combination of existing drugs and immunomodulatory compounds remains a promising strategy. For instance, the local administration of anti-TNF-α antibodies in combination with PM had shown a reduction in skin damage, although such a decrease did not correlate with parasite burden [31]. Similarly, the combination of PM and chloroquine during murine CL was able to decrease the lesion size but not the parasite load compared to data obtained from mice treated with the same doses of PM alone [31].

In the present study, the synthetic AMPs were able to diminish the number of intracellular macrophages when administered alone as well as in combination with drugs currently used in the clinic. They also increased the protective Th1-type immune response. Moreover, the peptide 19-4LF decreased the expression of genes related to Th2-type immune response cytokines in skin lesions, yielding a reduction in the parasite load. More studies are needed to completely elucidate the mechanism of action of such AMPs during *Leishmania* infection. Likely, they act through different targets including both the parasite elimination and the host immune response modulation.

## Figures and Tables

**Figure 1 pharmaceutics-14-02528-f001:**
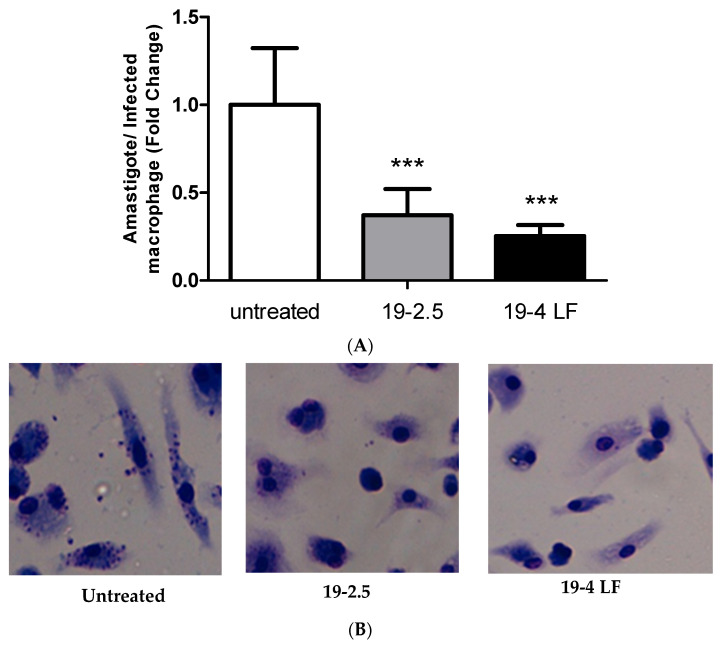
Leishmanicidal activity of 19-2.5 (1 µg/mL) and 19-4 LF (1 µg/mL) against intramacrophage amastigotes (**A**). Bars represent the mean ± SD values from three independent experiments. Significant reductions in the number of amastigotes per infected macrophage were detected when compared to untreated controls (***, *p* < 0.001). Representative images of untreated infected cells and infected macrophages treated with 1 µg/mL of 19-2.5 or 19-4LF peptides (**B**).

**Figure 2 pharmaceutics-14-02528-f002:**
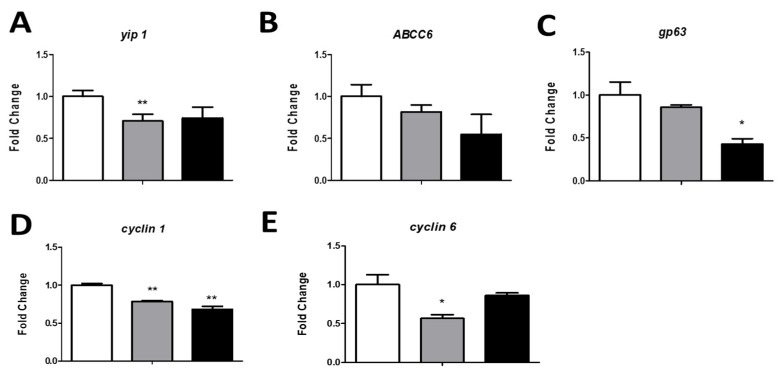
The effect of 19-2.5 and 19-4 LF on *L. major* amastigotes gene expression levels. The graphs show the expression levels of genes related to drug resistance, [*yip 1* (**A**), *ABCC6* (**B**)], virulence [*gp63* (**C**)], and *Leishmania* proliferation [*cyclin 1* (**D**), and *cyclin 6* (**E**)] after 24 h of treatment with 19-2.5 (1 µg/mL) and 19-4 LF (1 µg/mL). *GAPDH* was used as a housekeeping gene to normalize *L. major* gene expression. The amount of each transcript was expressed by the formula 2*^ct^*^(GAPDH)^^−^*^ct^*^(gene)^. Bars represent the mean ± SD from three independent experiments. Significant differences compared to control (untreated cells) were indicated (*, *p* < 0.05; **, *p* < 0.01).

**Figure 3 pharmaceutics-14-02528-f003:**
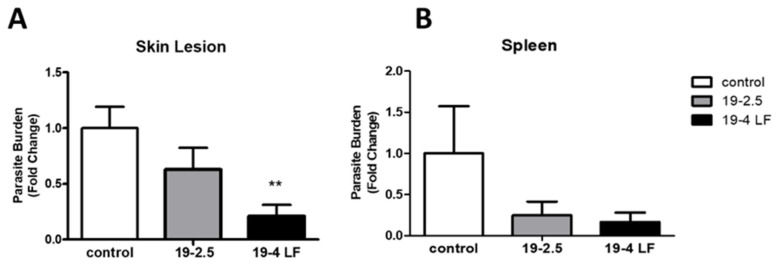
Effect of peptides 19-2.5 and 19-4 LF on parasite burden in the skin lesions (**A**) and the spleen (**B**) of *L. major* infected mice after treatment compared to controls (untreated mice). The parasite burden was evaluated by quantifying the *Lm18S* mRNA gene expression levels using qPCR method. Bars represent the mean ± the SD. Significant difference was indicated (**, *p* < 0.01 compared with the untreated control).

**Figure 4 pharmaceutics-14-02528-f004:**
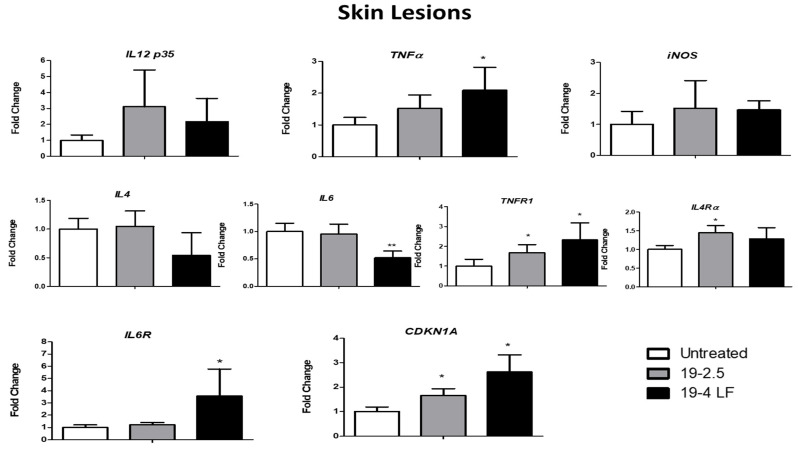
Gene expression levels of Th1 cytokines (*IL 12 p35* and *TNFα*), *iNOS* gene, Th2 cytokines (*IL 4* and *IL 6*), cytokine receptors (*TNFR1*, *IL4 Rα* and *IL6 R*) and *CDKN1A* gene from the skin lesions of BALB/c mice treated with 19-2.5 and 19-4 LF peptides or untreated. *β-actin* was used as a reference gene to normalize mouse gene expression. The amount of each transcript was expressed by the formula 2*^ct^*^(actin)−*ct*(gene)^. Bars represent the means (±SD) (*, *p* < 0.05; **, *p* < 0.01).

**Figure 5 pharmaceutics-14-02528-f005:**
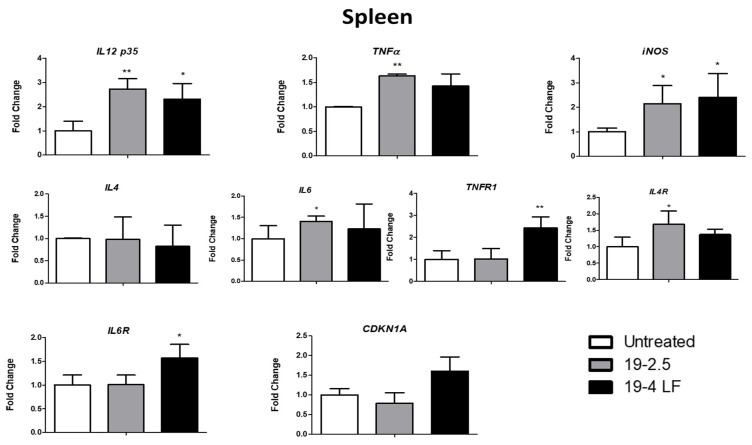
Gene expression levels of Th1 cytokines (*IL 12 p35* and *TNFα*), *iNOS* gene, Th2 cytokines (*IL 4* and *IL 6*), cytokine receptors (*TNFR1*, *IL4 Rα* and *IL6 R*) and *CDKN1A* gene from the spleen of BALB/c mice treated with 19-2.5 and 19-4 LF peptides or untreated. *β-actin* was used as a housekeeping gene to normalize mouse gene expression. The amount of each transcript was expressed by the formula 2*^ct^*^(actin)−*ct*(gene)^. Bars represent the means (±SD) (*, *p* < 0.05; **, *p* < 0.01).

**Figure 6 pharmaceutics-14-02528-f006:**
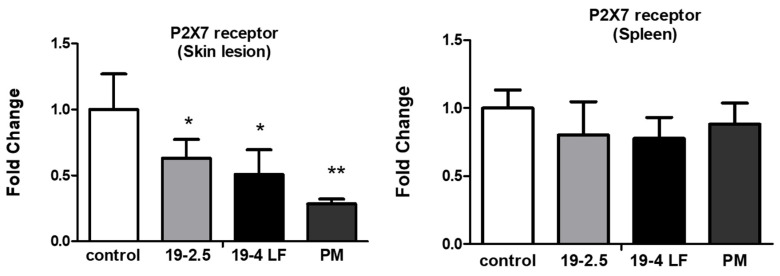
Effect of PM and the two peptides (19-2.5 and 19-4LF) on the expression levels of *P2X7 receptor* gene in the skin lesion and the spleen of treated BALB/c mice compared to untreated mice (controls). The bars represent the means ± SD. Significant differences were indicated (*, *p* < 0.05; **, *p* < 0.01 compared to controls).

**Figure 7 pharmaceutics-14-02528-f007:**
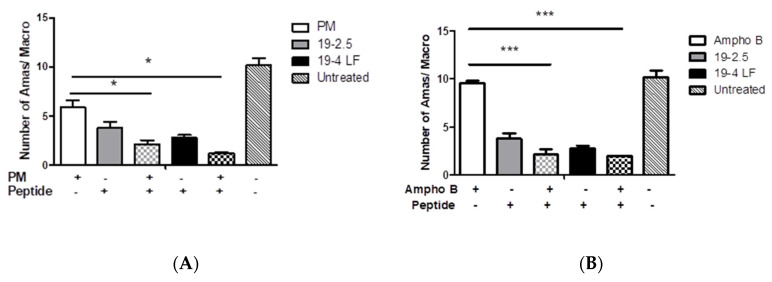
Leishmanicidal effect of PM (50 µM) (**A**), Ampho B (0.025 µM) (**B**) and their combinations with peptides Pep19-2.5 (1 µg/mL) and Pep19-4 LF (1 µg/mL). Bars represent the means (±SD) from three independent experiments (*, *p* < 0.05; ***, *p* < 0.001).

**Table 1 pharmaceutics-14-02528-t001:** Primer sequences used for the quantification of *Leishmania* genes.

Gene	Sense Primer (5′→3′)	Antisense Primer (5′→3′)
*Yip1*	AAGCTCCTTGGCAGCAAGAT	TGTGTCGGAAAAAGCGCAAG
*ABCC6*	TGTCCTCTCAACACGCATCC	TCGCAGAGCTCTTCAGTTGG
*gp63*	ACTGCCCGTTTGTTATCGAC	CCGGCGTACGACTTGACTAT
*Cyclin 1*	CCCCAACACCGCTGACTAAT	TCCGACTGGCGGTCTATGTA
*Cyclin 6*	AGTACCCTGCACGCCTACTA	TTGTTGTTGGCGCAGGAAAG
*Lm18S*	CCAAAGTGTGGAGATCGAAG	GGCCGGTAAAGGCCGAATAG
*GAPDH*	ACCACCATCCACTCCTACA	CGTGCTCGGGATGATGTTTA

**Table 2 pharmaceutics-14-02528-t002:** Primer sequences used for the quantification of mouse genes.

Gene	Sense Primer (5′→3′)	Antisense Primer (5′→3′)
*IL12p35*	CACGCTACCTCCTCTTTTTG	AGGCAACTCTCGTTCTTGTG
*TNFα*	CTTCCAGAACTCCAGGCGGT	GGTTTGCTACGACGTGGG
*iNOS*	TCCTACACCACACCAAACTG	AATCTCTGCCTATCCGTCTC
*IL4*	GCTATTGATGGGTCTCAACC	TCTGTGGTGTTCTTGTTGC
*IL6*	ACAAAGCCAGAGTCCTTCAG	TGGATGGTCTTGGTCCTTAG
*TNFR1*	CGATAAAGCCACACCCACAA	ACCTTTGCCCACTTTTCACC
*IL4Rα*	TGACCTACAAGGAACCCAGGC	GAACAGGCAAAACAACGGGAT
*IL6R*	GGAGATCCTGGAGGGTGACA	CGTTGTGGCTGGACTTGCTT
*CDKN1A*	TTGTCGCTGTCTTGCACTCT	GGCACTTCAGGGTTTTCTC
*β-actin*	CGCGTCCACCCGCGAG	CCTGGTGCCTAGGGCG

**Table 3 pharmaceutics-14-02528-t003:** Drug oral bioavailability parameters of 19-2.5 and 19-4 LF peptides.

Peptide.	LogP (≤5)	TPSA	MW (≤500)	nON (≤10)	nOHNH (≤5)	Vol (Å^3^)	Linpinski’s Violation (≤1)
19-2.5	−5.27	999.61	2712.28	59	48	2500.49	3
19-4LF	−5.28	938.93	2463.03	55	45	2316.94	3

LogP, logarithm of compound partition coefficient between n-octanol and water. TPSA, topological polar surface area. MW, molecular weight. nON, number of hydrogen bond acceptors, nOHNH, number of hydrogen bond donors.

## Data Availability

Not applicable.

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
