# Peer review of "Repurposing the Antibacterial Agents Peptide 19-4LF and Peptide 19-2.5 for Treatment of Cutaneous Leishmaniasis"

_pharmaceutics, 2022, doi:10.3390/pharmaceutics14112528_

Round 1

Reviewer 1 Report

Title: Repurposing the antibacterial agents peptide 19-4LF and Peptide 19-2.5 for treatment of cutaneous Leishmaniasis

Overview: El-Dirany and colleges present an article whose aim of the work was to repurposing AMPs as antileishmanial drugs.

The manuscript writing is well organized and the results are very interesting.

Minor revision: 

Check the entire text as italics are missing from the species names in several parts.

Material and Methods:

2.2 – Why used female mice ?

2.3 – The use of thioglycolate before obtaining peritoneal macrophages can not preactivate macrophages?

2.4 - Why was cytotoxicity only performed on bone marrow-derived macrophages if the infection experiments were performed on peritoneal macrophages?

2.5 -  Why such a high amount of parasite/cell (20/1) ? Haven't other quantities been tested?

Results

3.1 - I don't understand why the macrophage type was changed for the infection assays? It would be good to also show cytotoxicity on peritoneal macrophages.

3.2 - Figure captions must be self-explanatory.

In the legend of Figure 2, include which housekeeping was used and how the relative expression was calculated.

Adjust the table caption.

3.7 - Which macrophage was used?

The figure 7 should be show as the number of amastigotes/cells.

Author Response

Thank you.

Reviewer 2 Report

Antimicrobial peptides are diverse molecules, which provides multidimensional opportunities in different directions of drug discovery.  Tropical diseases, such as Leishmaniasis, remain underexplored. In agreement with this, databases reveal a large number of peptides with antibacterial and antitumor properties, but few peptides have been evaluated against Leishmania. Furthermore, most of these molecules have been studied using promastigotes and in vitro assays. The present study presents significant advances in the application of peptides in the treatment of Leishmania. The authors explored intracellular mechanisms, poorly explored in previous investigations. The manuscript is well written and clearly illustrates the potential of peptide repurposing. Below I list some suggestions.

1. Lines 28, 29, 33 ( and so on). Scientific names are always italicized.

2. The numbers are repeated in the bibliographic references.

3. Updated epidemiological data give a more realistic view of the impact of the disease. Update reference 1.

4. Line 128. References must be presented in order in square brackets.

5. Authors should include a microscopic image representative of the findings found in Figure 1.

6. Figure 1. The peptides did not completely reduce the infection of macrophages by the parasite. Is the same observed for the reference drug? Did the authors not include a positive control in this experiment?

7. Did the authors record the size of the lesion after treatment?

8. How many animals were used in the experiments?

Author Response

Thanks.
